# Development of an Ecologically Valid Assessment for Social Cognition Based on Real Interaction: Preliminary Results

**DOI:** 10.3390/bs12020054

**Published:** 2022-02-18

**Authors:** Guillermo Benito-Ruiz, Cristina Luzón-Collado, Javier Arrillaga-González, Guillermo Lahera

**Affiliations:** 1Faculty of Health Sciences, Isabel I University, 09003 Burgos, Spain; gbenitoruiz@gmail.com; 2Ohana Psychology Center, 28042 Madrid, Spain; cristina.luzon.co@gmail.com (C.L.-C.); javierarrillaga1986@gmail.com (J.A.-G.); 3Príncipe de Asturias University Hospital, 28805 Alcalá de Henares, Spain; 4Department of Medicine and Medical Specialties, University of Alcalá—IRyCIS, 28801 Alcalá de Henares, Spain; 5CIBERSAM (Biomedical Research Networking Centre in Mental Health), 28806 Alcalá de Henares, Spain

**Keywords:** social cognition, emotional intelligence, emotion recognition, theory of mind

## Abstract

Many social cognitive assessment measures that are appropriate for clinical use are currently available, but there is a general concern about their ecological validity. This study aimed to develop an applicable real interaction-based test to assess social cognition. A sample of 50 subjects (mean age 22 ± 5.8, 56% women) took the Social Interaction Test as well as two instruments for assessing social cognition: (1) the Movie for Assessment of Social Cognition (MASC) and (2) branch 4 from the Mayer–Salovey–Caruso Emotional Intelligence Test (MSCEIT). The test showed no incidence on its application. The reliability of the 18-item final version of the test was a medium-high level (Cronbach’s alpha = 0.701). To assess the internal structure of the test, a multidimensional scaling procedure was used. The common space of coordinates for the two-dimensional solution showed a normalized raw stress of 0.076 and Tucker’s congruence coefficient of 0.965. The social interaction test showed stronger association with MASC (more realistic, video-based format) than with MSCEIT (less realistic, paper-based format). The Social Interaction Test is applicable and feasible to use it to assess social cognition in the general population.

## 1. Introduction

Social cognition is the basic ability of the subject to adapt to the social environment. It enables the processing of social information and all anticipatory or subsequent cognitive acts in response to perceived social stimuli (such as emotions, intentions, double meanings, irony, etc.). Social cognition components are a permanent topic of discussion [1,2,3,4]. A well-known proposal of its components includes emotion processing, social perception, theory of mind/mental state attribution, and attributional style/bias [5]. The construct of social cognition is rather like an umbrella that encompasses different dimensions, though there is still an open debate on identifying which domains should be included [6,7,8,9,10,11].

In people whose social cognition is altered, there is a difficulty in perceiving and processing relevant stimuli to guide their interactions with other people observed. This impact on social cognition is not necessarily accompanied by poorer performance in other cognitive areas, suggesting that an altered ability in perceiving and processing a particular type of social information is sustained by specific regions of the brain [6]. 

The assessment of social cognition is a matter of concern since the currently available tests only value some of its components and often resort to unhelpful stimuli from an ecological point of view [12,13,14]. 

Ecological validity can be considered as the generalizability (veridicality, or the extent to which assessment results relate to and/or predict behaviors outside the test environment) and representativeness (verisimilitude, or the degree to which assessments resemble everyday life contexts in which the behaviors will be needed) [15]. Psychological assessment (not only) in social cognition has historically compromised ecological validity [16,17]. The differences between real social stimuli and those used in evaluation tools are clear enough for researchers to consider the possibility that these types of stimuli are processed by different psychological functions. Mesulam pointed out that cognitively impaired patients sometimes showed a good overall performance on assessment test batteries but failed on easier daily tasks related to those cognitive domains [18,19]. This assumption would lead to an admit mismatch between performance on tests and in “real-life situations” [20]. Other studies involving similar situations in neurocognitive assessments can illustrate this point of view [21]. This differences in outcomes outside the assessment setting is the main concern about ecological validity, since there is not a direct way to predict real-life functioning from traditional tools.

There are many tests available for social cognition that have proved their reliability, but they do not show convergence between them; scores in some tasks do not predict performance in others [20]. An analysis of the tools available to assess social cognition shows important differences between real-life social stimuli and those present in psychometric tests [22,23,24]. It cannot be assumed that the stimuli used to elicit cognitive processing in these tests are equivalent to “genuine” real social stimuli. Most of this assessment tools are available in a “paper and pencil” format, based on texts that recreate social vignettes from which the examined persons must make decisions. Some of the most commonly used tests of this kind are the Schema Component Sequencing Task-Revised [25], Situational Feature Recognition Test Version 2 (SFRT) [26], Ambiguous Intentions Hostility Questionnaire (AIHQ) [27], Profile of Nonverbal Sensitivity (PONS) [28], Interpersonal Perception Task (IPT) [29], Social Knowledge Scale (SKS) [30], Awareness of Social Inferences Test Part III [31], Bell Lysaker Emotion Recognition Test [32], Penn Emotion Recognition Test (ER-40) [33], Reading the Mind in the Eyes Test [34], and Hinting Task [35,36,37,38]. Available tests show high variability in the processes that they measure and, more importantly, in the degree of mental state representation needed to answer them [20].

Apart from real social stimuli, another important condition that is compromised in social cognition tests is the passive role of the assessed person. This is an important problem regarding to the mismatch of social cognition tests and real-life performance from our perspective. Not taking part of the social interaction in which stimuli are naturally involved, can lead the subject to perceive and process information in a different way than having an active role. Some available tests that imply certain kind of real interaction introduce important advances on ecological validity at this point. That is the case of a well-known test for theory of mind (the Director’s Task) [39,40]. It can be a more ecologically valid tool, since evaluated subjects need to take a second person’s perspective into account and perform actions, but those are oriented only to the declarative knowledge of other’s perspective (excluding perception and processing of emotions and intentions). An important distinction on some assessment tasks should be made between declarative and emotional knowledge since the construct ToM is used in different contexts. Perceiving and processing emotions and intentions of others (emotional ToM) is different from representing the other’s gaze, perspective, or non-emotional aspects (declarative ToM) [20]. This study focuses on the first ToM sense aspects (related to emotional intelligence, attributional style, and social perception rather than declarative knowledge).

All the previously mentioned tests show different psychometric profiles, but most of them lack a strong background for a deep analysis of reliability, validity, and scale [38]. The Mayer–Salovey–Caruso Emotional Intelligence Test (MSCEIT) can be considered to have a strong psychometric background, using a big sample and showing high reliability and discriminant validity [41]. The emotional management branch of the MSCEIT was proposed by the MATRICS consensus groups as the gold standard for the assessment of social cognition in schizophrenia [42,43,44]. In any case, differences of social cognition components and emotional intelligence are clear enough to consider them different constructs [45]. 

Reviews on this topic show opportunities for assessment in psychiatric settings [38], but its relevance to a subject’s general adaptative social functioning makes it a major topic for developmental, social, cognitive, and neuroscientific areas. 

This project aims to develop a test based on genuine social stimuli and allowing for real interaction to ensure representativeness from an ecological validity perspective. To do so, a controlled social situation was created so participants could be exposed to real social stimuli while participating in it. This test intends to use stimuli that are equivalent to those in social interactions, so its processing would rely on the same cognitive functions. First, we aimed to develop a real social cognition interaction-based test that can be used to assess a sample in an ecologically valid way. A basic condition for a tool of this kind is that it can be applicable without incidences and remaining natural (based on a naturalistic social interaction). In the second stage, we analyzed data exploring the validity and reliability of this new instrument.

## 2. Materials and Methods

### 2.1. Participants

We issued publicity advertisements and performed public speeches to inform university students and workers about a project related to cognitive research and asked for participants. Participation consisted of a 90-min psychological examination process. Participants were not offered compensation for their cooperation.

The recruited sample comprises a group of 50 subjects, with an average age of 22.7 years old (SD = 5.81) (56% women). All the subjects were recruited from a university setting; while most of them were undergraduate students, some were postdoctoral researchers. Subjects were interviewed individually by a psychologist from the research team before the assessment to screen those with any kind of existing psychiatric diagnosis or other condition that could affect their cognitive capabilities. None were being treated with psychotherapy or drug therapy at the time. All of them spoke Spanish as their native language. A sample of subjects with no presumable deficiencies in social cognition was considered to facilitate a normal distribution. Doing so allows for the reliability analysis to show if this tool could detect small differences between subjects with no deficiencies in social cognition. 

### 2.2. Research Design

#### 2.2.1. Social Interaction Assessment Development

The first task in developing the assessment tool was to identify the sub-variables that it should include. A meeting sponsored by the National Institute of Mental Health initially defined five domains of social cognition: emotion processing, theory of mind (ToM), attributional bias, social perception, and social knowledge [6]. According to this proposal, a set of 30 items was developed corresponding to these five domains as follows: emotional processing, 6 items; ToM, 10 items; attributional style, 6 items; social perception, 5 items; and social knowledge, 3 items. These items consisted of short, independent social interactions (i.e., vignettes) to be performed or recreated by two persons while interacting with the subjects, so that they could genuinely experience the social situation as participants of the interaction, not simply as external observers. These vignettes were designed to include the minimum stimuli required to assess the target domain, thus focusing only on a single social cognition aspect. The 30-item set was integrated into a broader social situation that linked all the items together. This consisted of a psychological examination with a fake cognitive test, in which participants had to cooperate with another subject under the instructions and supervision of an evaluator (both members of the researcher team). This fake test used during the evaluation consisted of a set of geometrical figures that had to be created from 12 different plastic pieces. The participants had to form the complex figures that the researcher showed to them, combining the plastic pieces by cooperating with the confederate subject in alternative turns. While this supposed cognitive evaluation was taking part, the members of the research team involved (evaluator and second participant) were able to perform every item of the test in the subject’s presence. Participants did not know while completing this task that the real interest of the study was assessing what they perceived from the social situation in which they were immersed. Hiding it was necessary to keep their attention unbiased toward social aspects of interest for the study.

A group of independent judges (AB, AF, AG, GL, LL, and MR) who were experienced psychiatrists and psychologists evaluated the script with the set of items. They answered a questionnaire that assessed every item on appropriateness, validity, and pertinence to its assumed social cognition component. For each domain (appropriateness, validity, and pertinence) in each item, not reaching a minimum of 8/10 in overall quality and/or an inter-judge variation of 25% (deviation quotient = 0.25) was sufficient to require reformulating that item. Eight items did not satisfy at least one of these conditions, so they were rewritten according to the recommendations provided by the judges. 

Two psychologists, who were members of the research team, rehearsed the script as the researcher and confederate subject at each administration of the test. An assessment tool with an inter-active component such as this required a great effort to reduce variations in the performance of the researchers involved in the recreated situation. A “performance” in this scenario must respect several conditions:-Natural feel: The actor’s performance must be accurate to the presented situation.-Credibility: Overall credibility, but most importantly in the expression of emotions, the performances corresponding to each item must be appropriate in intensity and type.-Rhythm: There must be extra natural interaction between items; otherwise, the resulting assessment could be perceived as too fast, “cold”, or hardly believable.

To create a credible interaction, actors were trained to introduce several fake test items during the assessment so that the overall feeling is the one to be expected in these kinds of tests.

Once the script was memorized by the researchers and all three credibility issues were addressed, the whole interaction was rehearsed until performed four times without any mistake. After that, a first administration of the test was conducted; a subject was recruited for this experience with the same conditions as that of the entire sample; thus, the experience was comparable to the real ones. In this “real-world scenario”, researchers were required to introduce every real item and some fake items to achieve credibility while adhering to the 20-minute length. It was considered to be long enough to allow for the items to be performed while keeping the subject’s attention. The independent group of judges watched a video recorded to evaluate the representation and each item of the test on the following variables:-Pertinence: The item could be included in this test (yes or no); there must be a consensus among all judges to keep the item in the test.-Relevance: The item was considered accurate if it assessed the cognitive domain that it was intended to assess (apparent validity) (5-point Likert scale).-Apparent discrimination: The item was useful to identify subjects with high and low cognitive capabilities in each domain (5-point Likert scale).

Items that met the following criteria based on the judges’ ratings were included as is in the final version of the test; otherwise, the items were revisited.

-All six judges consider the item to be pertinent.-The average relevance and apparent discrimination are 4 or higher.-The inter-judge deviation quotient on relevance and apparent discrimination is less than 0.25.

Eight items failed to meet these criteria, so they were redesigned and reevaluated before being included in the final version of the test to be administered to the sample. Once the new script was rehearsed again by the research team and a standardized performance was reached, the final version of the social interaction test was ready to be applied to the sample.

#### 2.2.2. Procedure

In the applied procedure, each of the 50 subjects of the sample were individually assigned to a separate room in a research laboratory in Madrid, Spain, equipped with cameras and microphones to complete a “spatial reasoning test” (the simulated assessment aforementioned). A member of the research team (who was introduced as the evaluator) explained the evaluation process to the subject. The (fake) test was presented as a cooperative spatial reasoning task, involving another participant “who was about to arrive”. The second participant was actually the other member of the research team, who had prepared the (real) evaluation test’s script. Since the very first moment in which the confederate arrived, the participant was taking part of the social situation containing the test’s items. The entire subject–confederates interaction was set to last for 20 min, so the evaluator had to include as many false items as necessary to reach that time length. The participant was informed that the entire test would be recorded on video for subsequent analysis of every detail of the administration. 

Once the social interaction was completed, another researcher entered the room and revealed the deception, explaining to the subject that they had not previously revealed the true intentions of this part of the assessment. In fact, the real test was about to start. The researcher team explained then to the participant that deception was needed to keep their attention free of any bias during the social interaction, letting them ask anything about the procedure. After that, the main researcher played the recorded video of the entire interaction on a monitor for each subject, pausing each time that an item was performed and then asking subjects about their perceptions and thoughts, noting that they should answer regarding the real experience, not the video playback. The questionnaire included the 30 items, in which participants were asked to verbally explain the mental states of the confederates during some moments of the interaction (including what they felt, thought, wanted, knew, or spectated). Compared to a list of answers considered correct that were developed for each of the 30 items, the researcher evaluated if the subject successfully identified the social cognition aspects regarding each item. Only in the case that the participant demonstrated complete awareness of the aspects of social cognition involved in that part of the interaction was the item scored as correct. Every right answer added one point to the subject’s final score on the test, partial answers did not score. 

After the social interaction test, each subject completed two more social cognition measures explained below.

### 2.3. Measures

Every participant in this study completed the test developed for this project as well as two well-known instruments for assessing social cognition (1) the Movie for the Assessment of Social Cognition (MASC) test [46,47] and (2) the Emotional and social management areas from the Mayer–Salovey–Caruso Emotional Intelligence Test (MSCEIT) [44]. Figure 1 shows the whole flow.

The MASC is a 45-min audiovisual measure that evaluates mentalizing (the ability to represent other’s mental states) by showing interactions between different characters in a film. Throughout is duration, the film is stopped 46 times to ask participants about the emotions, thoughts, and intentions of its protagonists. The MASC provides a global score of mentalizing and subscales assessing different types of mentalizing errors (i.e., over mentalizing, under mentalizing, and no mentalizing) [46]. It was chosen for being considered the more ecologically valid standardized tool at the moment. 

The MSCEIT Branch 4 is considered a key measure for social cognition, although it was originally conceived as an emotional intelligence measure. It is a well-documented psychometric test with an important validation background for several languages [8,43]. Emotional intelligence represents the ability to not only monitor, recognize, and reason about one’s own and other people’s emotions, but also to use this emotional information to guide one’s thinking and actions [48]. In this work, according to the MATRICS committee, only two scales were used (emotional management and emotional relations), which are called the “emotional management branch”. The MATRICS committee describes this test as a “paper-and-pencil multiple-choice test that assesses how people manage their emotions” [43]. It was included due to being considered a reliable and valid test proposed as the better option for social cognition assessment by MATRICS, although its paper and pencil design limits its ecological validity.

## 3. Results

The application of the social interaction test showed no incidence as none of the evaluated participants did not detect that the test was a previously prepared interaction. All the participants completed the entirety of the test administration and understood the fact that the test’s real aim was not initially revealed, and no debriefing was needed. The whole sample continued the evaluation after the social interaction-based test was finished.

All statistical analyses were done with SPSS 16.0 software. The scores from the original version of the test had a mean of 17.38 points and a standard deviation of 3.66 points. The scores from the social cognition test were statistically analyzed according to the psychometric properties of the set of items. The initial test reliability (Cronbach’s alpha) for the 30-item version was 0.51. Reliability was evaluated by using discrimination coefficients (DCs). DC 1 and 2 were obtained for each of the 30 items. Comparing Cronbach’s alpha and the DC1 and DC2 coefficients, both procedures showed equivalent results and identified the same items as bad contributors to the final version of the test. Considering the overall alpha and DC coefficients, we looked for a combination of fewer items that would result in an increase in reliability. The set of items that yielded better reliability were selected. A group of 18 items was chosen to reach a Crombach’s alpha of 0.701. This 18-item solution was considered to have a good balance be-tween total reliability and the number of items on the test, although no social perception items were included in it. The following results refer to the 18-item set. The resulting mean and standard deviation are shown in Table 1. Men and women showed statistically significant differences in their mean social interaction test scores. 

To explore the internal consistency of the items, a multidimensional scaling procedure was used since the sample size did not allow for a more powerful analysis. Table 2 shows the wellness-of-fit stress measures for the model. The common space of coordinates for the two-dimensional solution showed a normalized raw stress score of 0.076 and Tucker’s congruence coefficient of 0.965. We can assume an accurate fit of the multidimensional scaling model to this sample (normalized raw stress score close to zero and explained dispersion and Tucker’s congruence coefficient close to one).

Figure 2 shows the spatial distribution of each item on the two-dimensional solution of the multidimensional scaling. The closer the items appear, the more related their scores are. Blue dots indicate attributional style, green dots indicate emotional processing, red dots indicate theory of mind, and black dots indicate social knowledge.

Kolmogorov–Smirnov (K-S) tests were conducted to assess whether every variable in the study showed a normal distribution. The MSCEIT dimensions, MASC total score, Social Interaction Test total score, and the emotional processing subdimension of the Social Interaction Test had normal distributions (non-significant K-S tests). The other subdimensions of the Social Interaction Test (ToM, social knowledge, attributional style) showed a non-normal distribution.

This correlation matrix (Table 3) shows the Pearson’s correlation coefficients for every pair of tests and their dimensions. It is a double-entry table to show the association between variables.

## 4. Discussion

The aim of this study was to develop a measurement for social cognition in which the assessed subject was taking part of the social situation, and therefore, exposed to real social stimuli. Such an assessment tool should be based on real social interaction and needs to hide its real intention from the evaluated subject to preserve the subject’s natural attention and processing of social information. None of the participants in the sample discovered that the evaluator and the confederate were performing a role play with a script, and its administration showed no incidences of any kind. Once the best combination of items was selected, the reliability was a medium-high level, which suggests that instruments like this can be used for research purposes. The data obtained in this sample lead us to conclude that the Social Interaction Test is an applicable and reliable assessment tool; therefore, it is feasible to use for assessing social cognition in the general population. 

The 18-item combination used for the analysis excludes one of the five dimensions originally included in the design of the test (social perception). Thus, the hypothesis of the inner structure of this tool is threatened, as this dimension is not relevant to the overall reliability of the test. This can be due in part to bad discrimination between perception and processing of emotions in the original design of items, or (as is more likely the case considering the discrimination quotients) those items were so difficult that subjects could not effectively discriminate between them.

Men showed higher scores in the Social Interaction Test than women in this sample. The opposite has been reported in previous research [49]. The higher proportion of (declarative knowledge, non-emotional) ToM items and the lack of any item related to social perception in this tool (in which women show better performance) can explain the difference found in this sample. 

The association of the selected items, explored by a statistical technique of multidimensional scaling, partly showed that items of some of the sub-domains considered in the design of the questionnaire are related. These data only partially support the proposed structure of social cognition sub-scales in the test. Considering the poor consensus in the scientific literature related to the domains and validity of social cognition [20,50], this question requires more specific research and could not be clarified with the data obtained in this study. 

The correlation matrix showed a significant and moderate connection of the presented tool with the MASC, but no significant association with any of the MSCEIT scales or its composite score. The associations between these three different measures of social cognition are congruent with the ecological validity paradigm assumed. The differences between video and real interactions are too large to consider that tests based on each kind of stimuli are evaluating the same construct. Real social interactions are closer to video than to paper-and-pencil stimuli. This work sheds some light on this complex issue by showing a differential correlation between tests with different levels of ecological validity. 

Linking social cognition evaluation to real-world scenarios is shown to be problematic, as the construct complexity, the large variety of tools available, and their unclear relationships difficult this integration [5,20,50]. Linking social cognition domains to their biological correlates in naturalistic settings seems difficult to achieve now [51]. An ecologically valid assessment is then challenging since there are no available tools that can be considered based on “valid” stimuli or the subject’s performance. 

There are several limitations that should be taken into consideration. This study was oriented to test whether it is possible to develop an assessment tool based on real social situations for social cognition evaluation purposes. At this stage of the research, checking the applicability of the test was the main goal, so other aspects were conditioned to it in the design of the study. The fact that the participants reported their own cognitive processing once the interaction was completed can distort real outcomes, and some subjects could reorientate their answers during the interview while watching the video recording. The small size of the sample did not allow us to achieve statistical power on the psychometric analyses. Only the test’s applicability and reliability can be considered solid in this stage. 

This research should be considered preliminary rather than final. Experimenting with other samples of different age, cultural, educational, and clinical profiles will have the potential to improve the reliability and internal consistency. A larger sample is needed to extract factors and conduct a comparison of its unidimensional and/or multidimensional structure in order to explore internal consistency. Future research using tools based on this procedure could also explore associations between other social cognition assessment tools with different tasks and personal perspectives.

## Figures and Tables

**Figure 1 behavsci-12-00054-f001:**
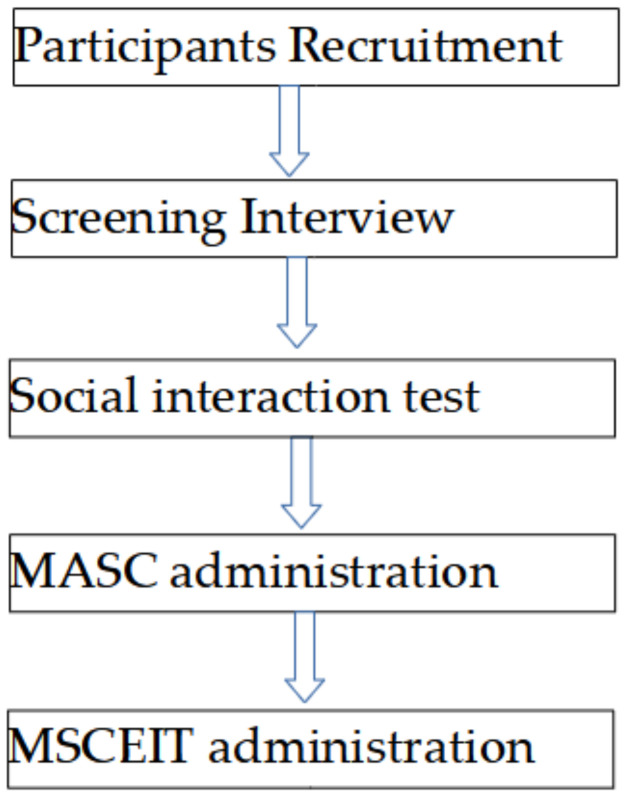
Flow chart of the assessment tools used.

**Figure 2 behavsci-12-00054-f002:**
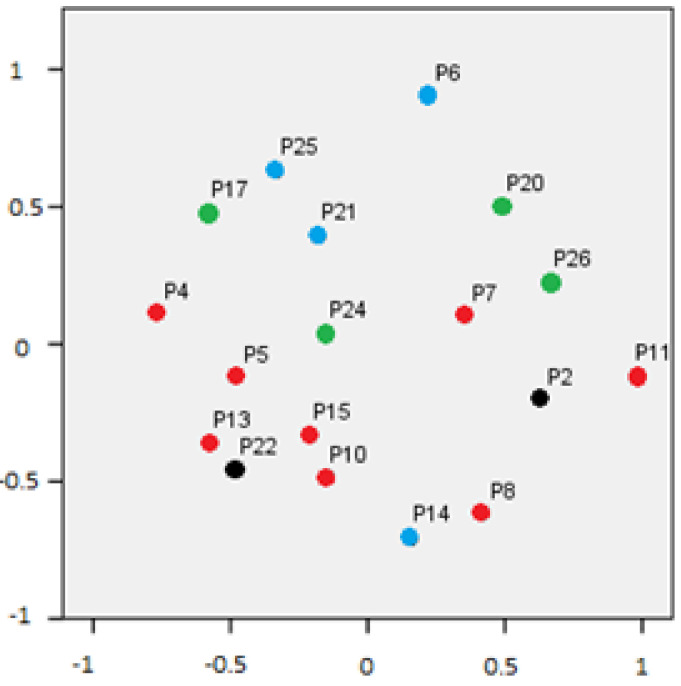
Two-dimension distribution of items.

**Table 1 behavsci-12-00054-t001:** Mean, standard deviations, and maximum and minimum scores of the 18-item version of the test.

	Total (*n* = 50)	Women (*n* = 28)	Men (*n* = 22)
Mean	10.02	8.96	11.36
Standard Deviation	3.47	3.49	3.03
Minimum	0	0	6
Maximum	18	15	18
F = 6.521 (Significance = 0.014) KS = 0.854 (Significance = 0.460) Crombach’s alpha = 0.701			

**Table 2 behavsci-12-00054-t002:** Wellness-of-fit measutes of the multidimensional scaling model.

Raw Standardized Stress	0.067
Stress-I	0.259 (*)
Stress-II	0.667 (*)
S-Stress	0.165 (**)
Dispersion accounted for	0.932
Tucker’s Congruence Coefficient	0.965

* Optimal scaling factor = 1072. ** Optimal scaling factor = 0.948.

**Table 3 behavsci-12-00054-t003:** Pearson Correlation matrix of the social cognition assessment tools.

	MSCEIT Emotional Management	MSCEIT Social Management	MSCEIT Emotional Management Branch	MASC Total Score	Social Interaction Test Score
MSCEIT Emotional Management	1				
MSCEIT Social Management	0.304 (*)	1			
MSCEIT Emotional Management Branch	0.694 (**)	0.882 (**)	1		
MASC total score	0.055	0.159	0.137	1	
Social Interaction Test Score	0.218	0.008	0.106	0.465 (**)	1

(Pearson’s correlation coefficients) (* *p* < 0.05; ** *p* < 0.01).

## Data Availability

The data presented in this study are available on request from the corresponding author. The data are not publicly available due to privacy.

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
