# Peer review of "Development of an Ecologically Valid Assessment for Social Cognition Based on Real Interaction: Preliminary Results"

_behavsci, 2022, doi:10.3390/bs12020054_

Round 1

Reviewer 1 Report

The authors tried to compare and contrast the validity of the assessment in two conditions. The goal of the study is stated very clearly, but the study procedure should be presented with more explicit information. 

  1. What is the conceptual definition of ecological validity in the paper? The authors stated several times about the importance of ecological validity, but its conceptual definition was not offered in the paper. To me, the authors attempted to evaluate the measurement validity of the assessment.
  2. Is this an experiment? It is very confusing about the procedure. Did all the 50 participants complete "The MASC is a forty-five-minute audiovisual measure," and perform the overall social situation prepared by the research team in which the 30 items were performed lasted for 190 20 minutes? If so, how did the authors handle the order effect?  Half of the participants should do the second test first to cancel off the order effect. 
  3. The operational definitions of the variable should be presented explicitly. The real interaction should be explained with a clearer description. It seems like each participant mainly observed the interaction between a researcher and a fake participant based on a script. If so, the interaction is similar to a video stimulus. If not, a dyadic effect between a participant with the research team is not controlled, so this is also a problem. 
  4. The method should have a better structure with the sections of the participants, research design, procedure, and measures. 
  5. In terms of the results, what kind of program did the authors use? The results should be presented with more acceptable statistical outcomes.  If the authors believed the scale is multidimensional, it should be tested against a unidimensional model, for instance. 
  6.  In order to evaluate the validity of the study, the results need more information, so I won't evaluate the discussion at this point. 

Author Response

REVIEWER 1

Thank you for your comments. We look forward to responding in the best possible way to all your suggestions and contributions. We really appreciate your insights and contributions to this work

The authors tried to compare and contrast the validity of the assessment in two conditions. The goal of the study is stated very clearly, but the study procedure should be presented with more explicit information. 

  1. What is the conceptual definition of ecological validity in the paper? The authors stated several times about the importance of ecological validity, but its conceptual definition was not offered in the paper. To me, the authors attempted to evaluate the measurement validity of the assessment

We have included the definition in the Introduction section to clarify the concept, and it’s original reference Dawson, D. R., & Marcotte, T. D. (2017). Special issue on ecological validity and cognitive assessment. Neuropsychological rehabilitation27(5), 599–602. https://doi.org/10.1080/09602011.2017.1313379 .

Examples of some common social cognition phenomena are cited in parenthesis.

The main goal of this work was to state the development process of a new social cognition test (Social Interaction Test) meeting ecological validity criteria. Validity of the assessment is discussed in the conclusions section.

  1. Is this an experiment? It is very confusing about the procedure. Did all the 50 participants complete "The MASC is a forty-five-minute audiovisual measure," and perform the overall social situation prepared by the research team in which the 30 items were performed lasted for 190 20 minutes? If so, how did the authors handle the order effect?  Half of the participants should do the second test first to cancel off the order effect. 

Thank you for pointing this potentially confusing issue. The study is not really an experiment, the information collected was oriented to explore the test’s psychometric characteristics. We hope that the present description is clearer to that point now. The procedure section has been re-written.

All the 50 participants completed the following tests: Social Interaction Test, MASC and MSCEIT, in this order. Since the three assessments are oriented to social cognition and theory of mind, Social Interaction Test was administered first to keep the sample naive and not drive their attention to crucial social and emotional aspects of the interaction. Order effect (if any) could affect only MASC and MSCEIT tests.

A flow chart has been added at the end of the procedure section

  1. The operational definitions of the variable should be presented explicitly. The real interaction should be explained with a clearer description. It seems like each participant mainly observed the interaction between a researcher and a fake participant based on a script. If so, the interaction is similar to a video stimulus. If not, a dyadic effect between a participant with the research team is not controlled, so this is also a problem. 

An operational definition of social cognition is included at the beginning of the introduction section. Examples and components are also included.

Regarding the real interaction test, each participant was involved in a real social interaction with two members of the research team. After the interaction was completed (and revealed the deception) the participant was asked about the social cognition items of the test (while watching the recorded interaction just as a reminder).

Indeed, the dyadic effect is present, but we think that it is reduced to some extent, since the interaction involves a (fake) researcher and other (fake) assessed subject and the items of interest for the study are related to both of them. The social interaction was designed and rehearsed with the objective to perform a standard set of stimuli to every participant

  1. The method should have a better structure with the sections of the participants, research design, procedure, and measures. 

Method has been rewritten following that structure

  1. In terms of the results, what kind of program did the authors use? The results should be presented with more acceptable statistical outcomes.  If the authors believed the scale is multidimensional, it should be tested against a unidimensional model, for instance.

SPSS 16.0 was used; it is now included in the results section.

We consider applicability and reliability the most important results of this study. A small sample like this does not allow more powerful contrasts (a factorial analyses would be useful to know the inner structure and dimensionality of the test). We consider the analysis as exploratory in that area, due to the small sample. The conclusions section includes this now as a limitation.

  1. In order to evaluate the validity of the study, the results need more information, so I won't evaluate the discussion at this point. 

The limited conclusions about validity are now clearer in the text in the conclusions section. It is only possible to consider that this data suggest that this procedure is closer to the more ecologically valid standard than to the non-ecological valid. We consider it a limitation in the conclusions section.

Reviewer 2 Report

The authors propose the Social Interaction Test, a new measure to assess social cognition. While the field of mentalizing/theory of mind/social cognition counts with possibly too many measurements, many of which require further proper psychometric assessment (e.g., Quesque & Rosetti, 2020), the goal of creating a more ecologically valid measure of social cognition provides an interesting contribution to the field. However, there are a number of concerns regarding the methodology, analyses, and goals of the study that I describe in detail below. Specifically, the purpose of the study does not seem to match the analyses or the findings; the background research used to introduce the rationale of the study lacks a proper review of previous literature. The methodology, especially what it concerns to the choice of participants and the development of the task, is not sound. Finally, the analyses used do not seem to address the question of whether the task proposed is an adequate measure of social cognition. Overall, it is unclear that the authors achieved their goal of creating an ecologically valid task of social cognition that measures most of its subcomponents and that involves a “real life” situation.

Recommended action: reject.

Bolded comments indicate major issues.

Main comments

            Abstract

  • The application of the test showed no incidence or side effects
  • What do the authors mean by side effects            

            Introduction

  • The authors mention: “the assessment of social cognition is a matter of concern since the currently available tests only value some of its components and often resort to unhelpful stimuli from an ecological point of view [10-12].”
  • What do the authors mean by unhelpful stimuli from a psychometric perspective? The authors should unpack this point, given that there are dozens of tests of social cognition, emotional intelligence, theory of mind, and other constructs currently used by researchers to measure social cognition to some degree. The validity of some of these measures has been supported in multiple studies, while other tests seem to present poorer validity (therefore there is variability among the measures). Importantly, many of these tests are indicators of real-life outcomes (predictive validity), correlate with other cognitive and noncognitive performance measures (convergent and divergent validity), and are even used as clinical tools to assess social-cognitive disorders. A more thorough review of why current available tests fail to meet ecological validity standards should be conducted, with an emphasis on why it is important to create more ecologically valid tests and how to overcome their lack of construct validity and reliability. This would provide a stronger argument for the need of the Social Interaction Test.
  • In addition, the authors should discuss one famously ecologically valid task of mentalizing that provided a first-person measure of social cognition in adults by Keysar Lin and Bar (2002) that was later implemented on a computer-based task by Apperly and colleagues (Dumontheil, Apperly, & Blakemore, 2010). This task originally videotaped interactions between confederate-participants pairs on a number of social-cognitive and perspective-taking processes that seem to overlap with the goal of the authors.

  • Similarly, in pages 1-2, the authors mention: “These assessment tools are usually available in a “paper and pencil” format based on texts that recreate social vignettes from which the examined persons must make decisions”.
  • However, these tasks are not necessarily the most common or utilized today in all the areas that study social cognition. In fact, computer-based tasks, videos, eye-and mouse-tracking paradigms, and EEG-recording tasks are used often instead of the original paper and pencil tasks/vignettes, especially when social cognition is not measured in children. For a more thorough review of some, but not all, of the tasks available to date, see Quesque & Rosetti (2020).

  • The authors might also want to broaden the focus of literature review to research outside clinical-focus tasks and applications, as other researchers in developmental, social, cognitive, and neuroscientific areas study social cognitive constructs using multiple tasks (for example, Schaafsma et al., 2015; for a review, see Navarro, 2021, Unpublished Dissertation).

  • “but none of them provide a strong background for a deep analysis of reliability, validity, and scale”.
  • What do the authors mean exactly in this sentence? Some of these tasks have in fact already been included in studies where these properties are examined (e.g., Warnell & Redcay, 2019).

  • “This is not the case for the Emotional Management branch of the Mayer-Salovey- Caruso Emotional Intelligence Test (MSCEIT), which was proposed by the MATRICS consensus groups as the gold standard for the assessment of social cognition in schizophrenia [30-32].”
  • Similarly, please elaborate on why this task does meet the requirements for an exploration of psychometric properties.

  • In the current state of the manuscript, the authors fail to make a case for why none of the currently available tasks lack ecological validity. Is it the stimuli used in the tasks? Is it the lack of psychometric assessments? Is it the settings in which the tests are taken?

  • In addition, the researchers fail to consider the populations in which these tests are employed. While many of the myriad of tests currently available to assess aspects of social cognition can fall short of adequately assessing social cognition in neuro-atypical populations, they have shown to be adequate predictors of other cognitive and noncognitive abilities in typically developing children and adults (e.g., Diaz & Farrar, 2010, Dumontheil et al., 2016; Navarro & Conway, 2021; Samson, Apperly, Braithwaite, Andrews, & Bodley Scott, 2010; German & Hehman, 2016). The authors would need to clarify the focus the study is in clinical populations or focus their review on ecological validity in other samples as well.

  • The differences between real social stimuli and those used in evaluation tools are clear enough for researchers to consider the possibility that these types of stimuli are processed by different psychological functions.” This point made by researchers has indeed been made by multiple researchers to date (e.g., Schaafsma et al., 2015; Warnell & Redcay, 2019; Quesque & Rosetti, 2020) In fact, many researchers are currently focusing on describing the different aspects of social cognition rather than assuming that it embeds multiple social and cognitive subprocesses (for example, Apperly & Butterfill, 2009, 2010). The authors should review this past research and explain how their contribution adds to the search for the subprocesses that form social cognition. Importantly, most researchers would agree that using one task to measure social cognition is not an adequate representation of an individual’s social cognition, but rather multiple measures should be used.
  • The authors should describe how their measure complements an aspect of social cognition that hasn’t been properly assessed to date. In addition, a large number of tasks have been developed in the past 10-15 years, therefore more updated references are needed to provide an actual picture of the state of the literature.
  • This sentence isn’t entirely clear: “To do so, a controlled social situation was created so participants could be evaluated on their perceiving and processing capabilities such authentic social information.” In addition, please describe “perceiving and processing capabilities” in terms of operational definition.
  • so the processing of such information relies solely on social cognitive domains.” It is widely accepted that it is highly unlikely that a single task can be created to assess an entire construct both psychometrically and theoretically. Please, rephrase this sentence to better capture this issue.
  • Please, clarify “normal conditions” throughout.

Methods

  • What was the rationale for the sample size? Was a power analysis conducted a priori? If so, what was the power the authors aimed for? Please report this information.
  • In general, it seems problematic that a study that is aimed at validating a new test only recruits 50 participants, as the authors will be underpowered for most psychometric analyses.
  • Given the introduction, it seems that the sample chosen should rather be composed of clinical subjects rather than otherwise healthy university students. If the study aims to generalize the results of the study to a neuro-atypical sample, then why recruiting healthy subjects? If, as suggested above, the introduction includes literature on other areas and populations, it would be easier to generalize results.
  • small differences between subjects … and detect larger differences between subjects” I assume one of these was intended to be within-subjects.
  • Since the authors indicate that the goal of the study is to look at both between and within subjects effects, it is very likely that the sample size is not sufficient to achieve power. The authors should conduct a posterior power analysis.
  • Please provide information about the pre-screening interview (where did it happen?, who conducted it?, how long was it?, where the subjects interviewed individually?…).
  • What did subjects do in the 90 minutes meeting? Was this the duration of the whole study or just the pre-screening?
  • Why did subjects complete the two other measures (MASC and MSCEIT)? The authors don’t specify what was the purpose of these tasks of why they chose them. Both tasks seem to rely strongly on an emotional component of social cognition. Was there a rationale for this?
  • Given that all tasks were scored by asking participants about their impressions but no other behavioral measure was collected, the authors should note the limitations of asking participants to report on their own cognitive processing.
  • The authors only introduce Emotional Intelligence in the methods section. Given that the study is largely focused on the EI component of social cognition, they should introduce this concept in the introduction and explain why and how this subcomponent is relevant to their study and their proposed task.
  • Regarding the Social Interaction Test, the researchers critique early on their paper the use of vignettes and paper and pencil tasks as lacking ecological validity, but their own is essentially an acted vignette/video. Even if the vignette is acted out in a skit, it is very likely that a practiced skit does not elicit the same response in social cognition than a real situation would. If these skits and/or the subjects’ responses to the skits are available, it would be helpful to see them to understand if these skits are actually ecologically valid. Adding them to supplementary materials would be helpful.
  • In addition, Quesque & Rosetti (2020) point out that in mentalizing, there is a crucial difference between being part of the situation as a 1st person actor and as a 3rd person observer. There are already tasks that assess either one or both of these perspectives, however only 1st person tasks are considered to be telling of an individual’s actual social cognitive responses. If I understand the procedure correctly, this Social Interaction Test only assesses 3rd person responses.
  • The authors state “so that the subjects could genuinely experience the social situation as a participant not simply as an external observant.” Please explain how the subject was involved in the skit so that he was actively participating.
  • Relatedly, was the skit the same every time? Did it change based on subjects’ responses? How was the scenario introduced? Was there deception or did the participant know the interaction was planned? What responses (DVs) were collected? Please describe these and other aspects of the task in more detail at the beginning of the description of the task.
  • Please, explain who the experts were, what was their interrater reliability (cohen’s kappa) in addition to percentages and SD, and how they were chosen/how they assessed the task (were the skits performed in front of them/recorded?). Also, who performed the skits? Students? Research assistants? PIs?
  • Please explain what “false researcher” and “false assessed” mean in text.
  • Since one participant was a confederate, the researchers should explicitly say that deception was involved at the beginning of the Method. Also, please include information about how subjects were debriefed.
  • What is the distractor test? Why is there one and when is it administered?
  • I strongly recommend to build a Flow Chart for the method/procedure to explain when and how each element was introduced in the study, including:
  • Pre-screening
  • MASC and MSCEIT
  • Social Interaction Test
  • Distractor Test
  • The method section also needs a clarification of the elements of the Social Interaction Test:
  • Definitions of: False researcher, false assessed, false items, simulated assessment
  • Definitions of: real items are mentioned in page 4
  • Number of each type of item and number of skits (if more than one)
  • Randomization and counterbalancing of items/categories
  • Researchers involved in skit (including age, gender, rank, etc.) and whether they’re always the same.
  • What was the involvement of the participant; was he allowed to talk during the skit? How was this accounted for?

  • “so the false researcher had to include as many false items as necessary to reach that time length.” What does “false items” mean and what does “as necessary” mean here? Did the number of false items vary by participant? If so, why?
  • “After that, a new administration of the test was conducted.” Do the authors mean that a pilot study was first conducted? Or that a pilot was performed in front of the experts? This is not clear.
  • Similarly, I’m not clear what this sentence indicates: “This time, a subject was recruited for this experience with the same conditions as that of the entire sample; thus, the experience was comparable to the real ones.” Is this subject part of the 50-subject sample? Was another sample recruited for piloting?
  • “Once this first version of the test (items and representation) was completed, a group of experts on social cognition evaluated it to ensure validity and overall quality” Were any participants involved in this first version? Who examined their responses?
  • As mentioned, please report interrater reliability cohen’s kappa coefficients as well.
  • Were subjects aware of microphones and cameras and if so what were they told about them when they arrived?
  • A discussion of potential researcher bias effects should be included in the limitations of the study.
  • In page 4, the researchers reveal the DV of the study: “The subjects were asked to describe their perceptions and thoughts on some parts of the interaction while watching the recorded video, noting the importance of distinguishing between the real experience and the one while watching the video.” Psychometric properties aside, this methodology seems problematic. The subjects are after all watching a video of themselves and retroactively describing their emotions vaguely after the purpose of the study and the deception have been revealed. In addition, they now know the people involved in the skit are researchers which is likely to bias spoken responses. Methodologically, the responses to this task are of dubious quality.
  • The authors ask about perceptions and thoughts but they do not clarify what responses qualify as “correct”.
  • “Only in the case that the participant demonstrated complete awareness of the aspects of social cognition involved in that part of the interaction was the item scored as correct.” What does this mean in terms of operational definition and scoring?

Results

  • In line 210, how was the manipulation check assessed and at what point in the procedure?
  • First paragraph of results should go in Methods.
  • Re scoring: How were the items scored, was a correct answer a 1 and an incorrect a 0? Was partial score given in any case? Were scores standardized? What is the range of the scores for each item/subgroup of items?
  • What sample size are the results of Paragraph 2 based on? Authors say “original version of the test” - was this not the test taken by 50 subjects?
  • “The final selection of eighteen items had a reliability score of 0.701”. – do the authors mean Cronbach’s alpha reliability?
  • In line 229, authors say: “To assess the internal consistency of the items” I think they mean construct validity. Internal consistency refers to the reliability of the test, but this approach is thought to assess validity (whether the test measures what it is thought to measure).
  • Reliability estimates should be included in Table 1. Sample size per group and total should also be included in Table 1.
  • “Men and women showed statistically significant differences in their mean scores.”
  • The authors should hypothesize why this significant result was found. In addition, why is an F statistic reported for this difference and not a t statistic from comparing two means?
  • Indicate what KS statistic is in footnote or in text.
  • Explain what the multidimensional scaling procedure – MDS is before describing results.
  • It is unclear why the authors would use MDS to examine either internal consistency or validity. MDS is purely a visualization technique that presents items that are similar to each other and plots them. Because it is not rotated it does not provide information about how well the items in the test belong to the overarching construct they’re supposed to measure or the higher construct of social cognition. It just shows that some of these items are evidently related as they are thought to be.
  • However, in any case, the sample used in this study is too low to perform even the simplest psychometric assessment and obtain reliable results. A simple factor analysis would require upwards of 150 subjects. I recommend reading Kline (2015) and ter Braak (1994) for a thorough discussion on requirements for these kinds of analyses.
  • Another point of concern is that the analysis performed does not provide much information about the factors tapped by the items unless it uses some type of rotation method. This is likely why figure 1 does not present clustered factors.
  • In Figure 1 it is unclear why multiple items are missing of the full items the test is made of. It is possible that since this technique just shows the items that are related to each other using a cartesian distance, many of these items are outside of the plot boundaries.
  • The correlation table should be described and introduced before the rest of the analyses are performed. I also think the authors might want to present a diagonal correlation table unless there is a reason to present the full table.
  • I would like to know the authors’ thoughts on their task being largely solely correlated to the MASC task. Given that the test is thought to assess multiple components of social cognition but it only really correlated with MASC, is it possible that the nature of the correlation is due to both tasks involving verbal description of an observed situation on the same means (video)?
  • Overall, it is unclear what the purpose of the analyses was. Since the authors are presenting a new task, the purpose should be how well the items in the task group under the categories they are thought to measure (i.e., emotional processing, theory of mind …) using a measurement model first (confirmatory factor analysis - CFA) and then create a model in which all three tasks assessed (EI, MASC, Social Interaction test) are used to assess an overarching social cognition construct (i.e., hierarchical CFA) or even predict MASC and EI from Social Interaction task (i.e., Structural Equation Modeling). I’m not clear why how the analyses conducted show that the task is an adequate task of social cognition, nor how it shows it is ecologically valid, other than visualizing the tasks’ reliability. Importantly, a task being reliable (consistently providing the same results over time) doesn’t imply that a task that is valid (actually measures what it is intended to measure).

Discussion:

  • “the Social Interaction Test is applicable and well tolerated by a group of participants.”
  • It is not cleat what this means.
  • The authors imply that because the manipulation wasn’t discovered by the subjects, it indicates that the task adequately measures social cognition. That seems misleading. I think that sentence should be removed.
  • “The final 18 item combination excludes one of the five dimensions (emotional processing) originally included in the design of the test.”
  • This wasn’t stated in the Method or Results.
  • “Regarding the validity of the test, the correlation matrix showed a significant and moderate connection with the MASC, but no significant association with any of the MSCEIT scales. This confirmed the validity of the instrument hereby presented”
  • I don’t see how the lack of correlation with one task demonstrates validity. Or why ecological validity means lack of correlations. Importantly, correlations are not direct indicators of validity. Verbal fluency and Cognitive control are correlated but they do not assess the same construct. The authors suggest that the correlations of the tasks are indicators of the administration format rather than the underlying cognitive processes they share. This argument contradicts their inferences. If the tasks are merely correlated because of how they are administered, then they might not measure the same thing at all.

Minor:

  • English should be revised in some sentences for clarity.
  • Overall organization of the paper should be revised throughout. For example, the method section explains the task after much has been said about the task. The results section talks about descriptive statistics after the main analyses.
  • Headings levels are mismatched at times.
  • The terminology of the task is inconsistent and introduced throughout without proper introduction.
  • Typo in 186: to for
  • Authors don’t include deception debrief in their supplementary materials or explain it in text.

Author Response

Thank you for your comments. We look forward to responding in the best possible way to all your suggestions and contributions. We really appreciate your insights and contributions to this work

Reviewer 3 Report

COMMENTS TO THE AUTHORS

INTRODUCTION

  1. It should say social environment
  2. Name at least some of them. I suggest placing here the paragraph set at rows 115- 118
  3. Erase the hyphen of “nor-mal”

MATERIAL AND METHOD

  1. This subtitle should be in the same style as in the others (remove italics)
  2. How did the researchers operationalize this estimation?
  3. The concept of “mentalizing wasn’t explained previously in the introduction.

115 to 118. It will be better to place this paragraph early, in the introduction, given that defines social cognition and its components.

120 ToM: these sigla should be included the first time you mention Theory of Mind (ToM) if you will use it consistently through the text, not just once. Replace the comma with a semicolon before “attributional style”, which is another dimension.

124 “as a participant, not only” …place a comma here

182 Place a comma behind “Spain”.

RESULTS

210: Rephrase this sentence. Because it’s in another section, the use of “such” is incorrect. For example, there were no incidences on the application of the test, any participant detected, etc.)

215: The original version includes 30 or 18 items?

Table 1. I wonder if the results for men and women are correctly placed, given that most of the studies about sex differences in social cognition show higher general scores for women or no differences. If it’s correct, these results deserve some thinking about the possible reasons.

Table 3: It seems to be a Pearson’s correlation but should be specified in the title, given that there are other methods to estimate correlations. Also, it lacks a note explaining the significance of r coefficients (* and **)

DISCUSSION

  1. At what point was the emotional processing dimension cut out of the test? You should clarify if it was before the application or at the statistical analyses because there is a considerable difference between the original 30 items scale and the 18 items final version mentioned here.

CONCLUSIONS

313-314. If one of the main goals of this stage of the research were to evaluate the applicability of the test, you should mention it specifically in the goals’ paragraph (70 -78).

Author Response

(The authors gave the same response as above.)

Round 2

Reviewer 1 Report

Although the authors well addressed all the comments I mentioned in the previous review, I'm very skeptical about the news value of the study. The main goal of the study is about ecological validity and its definition stated that "Ecological validity can be considered as the generalizability." If so, 50 participants are not acceptable. If the study is about a rare case like a certain illness or uncommon event, then 50 participants would be best in a given situation.  However, the study used lay adults, so the number of 50 cannot be rationalized unless a strong rationale is given.   This journal is indexed SCI Q2. If the study has an exploratory approach with preliminary results, a more regional journals would be a good outlet.  

Author Response

We are glad to know that this version well addressed all the comments you mentioned in the previous review.

We share the main concern about small sample size, since it compromises the generalizability of the results and overall psychometric profile of this test. This has been pointed in the Discussion section as a relevant limitation, and we consider only applicability and reliability as solid psychometric aspects. We also conclude that a larger sample is needed to conduct stronger statistical contrasts. Internal consistency, validity analyses and correlation with other tools would undoubtedly improve with larger N value.

This work shows some important limitations on the available assessing tools on the field (even with the sample limitations) and opens a new way to develop new tools and procedures to overcome them. The tool described in this work is a modest but solid example of a new line of assessment in a difficult field, so more research can be done based on it. The publication of modest but well-directed work may lead to the development of useful social cognition assessment tools in the clinical setting.

Reviewer 2 Report

The authors addressed all my questions and concerns adequately and I consider that this version of the manuscript provides a better description of the intent and results of the study. 

Author Response

Thank you very much. We are glad to know that our new version addressed all your questions and concerns adequately. Your contribution to the final result has been very relevant.